# Digital Twin Models for Personalised and Predictive Medicine in Ophthalmology

**Miruna-Elena Iliuţă** [1,*], **Mihnea-Alexandru Moisescu** [1], **Simona-Iuliana Caramihai** [1], **Alexandra Cernian** [1], **Eugen Pop** [1], **Daniel-Ioan Chiş** [1] **and Traian-Costin Mitulescu** [2]

1   Faculty of Automatic Control and Computers, Automation and Industrial Informatics, National University of Science and Technology Politehnica Bucharest, 060042 Bucharest, Romania; mihnea.moisescu@upb.ro (M.-A.M.); caramihai2002@yahoo.com (S.-I.C.); alexandra.cernian@upb.ro (A.C.); eugen.pop@upb.ro (E.P.); daniel_ioan.chis@upb.ro (D.-I.C.)

2   Department 12—Ophthalmology, ENT, University of Medicine and Pharmacy Carol Davila Bucharest, 020021 Bucharest, Romania; costin.mitulescu@umfcd.ro

*   Correspondence: miruna_elena.iliuta@upb.ro

**Abstract:** This article explores the integration of Digital Twins in Systems and Predictive Medicine, focusing on eye diagnosis. By utilizing the Digital Twin models, the proposed framework can support early diagnosis and predict evolution after treatment by providing customized simulation scenarios. Furthermore, a structured architectural framework comprising five levels has been proposed, integrating Digital Twin, Systems Medicine, and Predictive Medicine for managing eye diseases. Based on demographic parameters, statistics were performed to identify potential correlations that may contribute to predispositions to glaucoma. With the aid of a dataset, a neural network was trained with the goal of identifying glaucoma. This comprehensive approach, based on statistical analysis and Machine Learning, is a promising method to enhance diagnostic accuracy and provide personalized treatment approaches.

**Keywords:** medical informatics; digital twin; predictive medicine; systems medicine; genetic eye diseases; glaucoma

## 1. Introduction

In the current medical context, according to a study conducted in recent years by the World Health Organization (WHO) [1], the number of individuals identified with visual impairments or blindness has significantly increased, reaching a global level of 2.2 billion, underscoring that ocular diseases pose a major challenge to the healthcare system.

Among the categories predisposed to the onset of eye diseases, individuals over the age of 50 stand out, and the causes leading to these impairments are represented by glaucoma, refractive errors, cataracts, diabetic retinopathy, and age-related macular degeneration. From an economic standpoint, regions with low and middle incomes have identified the most cases of visual impairments, both at a distance and nearby. According to [1], 80% of near-vision impairments have been identified in Western, Eastern, and Central sub-Saharan Africa—regions with low and middle incomes, while in North America, Australia, and Western Europe, less than 10% of cases were reported—regions with high incomes.

Based on the statistics presented in [1], one of the main causes of blindness is represented by glaucoma, with over 76 million people diagnosed with this condition in recent years. Due to the fact that glaucoma is asymptomatic in its early stages and progresses slowly, approximately half of the diagnosed individuals were unaware of the existence of this condition in their eyes. Thus, the WHO aims to conduct awareness and education campaigns for patients regarding the importance of eye care, and accessibility to healthcare services, thereby fostering the development of a patient-centered approach.

With rapid technological evolution, the world is in the midst of a process of continuous adjustment and change. By interconnecting the concepts of Digital Twin, Systems Medicine, and Predictive Medicine, revolutionary medical approaches can prioritize patients' health and safety.

The concept of a Digital Twin [2] originates from Michael Grieves' 2003 presentation on product life-cycle management based on his work with John Vickers [3]. Grieves and Vicker's motivation for developing the concept was to shift from the predominantly paper-based and manual product data to a digital model of the product which would become foundational for life-cycle management.

Systems Medicine [4] represents an innovative approach to understanding both health and disease, providing personalized insights into each individual's unique health journey across molecular, cellular, and organ levels.

Predictive Medicine [5] involves the proactive identification of changes in a patient's health condition before the onset of noticeable deterioration or improvement. Predictive Medicine also plays a crucial role in anticipating and managing therapy-related side effects, and optimizing patient care outcomes.

With the rapid evolution of technology and the world undergoing continuous transformation and change, we have set out to explore the use of Digital Twins in ophthalmology by creating an architectural framework that integrates datasets to make predictions regarding the identification of glaucoma. By integrating a dataset containing medical images and the results of ophthalmological examinations, along with information derived from the patient's history, glaucoma can be identified. Through the integration of Systems Medicine into the construction of a Digital Twin, genes that may influence predispositions to inherited pathologies can be detected, thereby contributing to increased accuracy in the provided results. To achieve this goal, we propose the utilization of Machine Learning to develop a model aimed at supporting early diagnosis and predicting the progression of glaucoma, thereby assisting clinicians in selecting the appropriate treatment.

Creating a Digital Twin associated with the patient contributes to monitoring the patient's health status and constructing personalized treatment plans [6]. Additionally, Systems Medicine and Predictive Medicine will identify various approaches to highlight complex interactions in the human body at different levels of organization [7]. Through DNA sequencing and the use of imaging protocols, biosensors, and wireless health monitoring devices, genetic mutations that lead to the onset of hereditary pathologies can be identified. Based on these results, the risk of each patient developing such pathologies can be evaluated [8]. Thus, through the use of genetic sequencing procedures and genetic tests, patients have the opportunity to analyze their DNA and identify any genetic mutations, which can provide information related to early diagnosis and the development of an optimal solution for personalized treatment.

The integration of Digital Twins, Systems Medicine, and Predictive Medicine in ophthalmology can lead to significant cost reduction by improving disease management, decreasing the need for costly medical interventions through early detection of eye diseases, and simulating the progression of pathologies. Another factor could be the optimization of treatment for each patient through predictive analyses, thereby reducing the risk of providing ineffective treatments. Furthermore, it should be mentioned that in the creation of the Digital Twin associated with the patient for the identification of glaucoma, ensuring the accuracy of the results requires the utilization of large volumes of data, and the process of collecting, processing, and transforming them into information is time-consuming. Additionally, issues related to the confidentiality and security of patient data, as well as the ethical risks that may arise, need to be addressed.

However, one of the limitations is the monitoring of the progression of pathologies because there are no sensors to capture real-time information from the patient, and monitoring can only be carried out with the help of a specialist doctor. Thus, monitoring can be performed by the specialist doctor at well-defined time intervals depending on the pathology. They will input the data obtained from the consultation regarding the patient's health

status, which will lead to the establishment of a human-to-machine (H2M) connection. Therefore, with the help of the Digital Twin, healthcare personnel are provided with an integrative approach regarding the identification of predispositions to certain pathologies, the early detection of these pathologies, treatment suggestions, the visualization of the stages of progression, and the patient's reactions to the proposed medication.

The paper is structured as follows: the specialized literature discusses selected definitions for the concepts of Digital Twin, Systems Medicine, and Predictive Medicine. (Section 2.1), respectively, in ophthalmology (Section 2.2), highlighting their potential benefits and relevance in this field. Section 3.1 introduces an architectural framework for Digital Twin in ophthalmology, while Section 3.2 focuses on identifying risks and assessing vulnerability in the context of Digital Twin. Subsequently, in Section 4, the datasets used for statistical analysis (Section 4.1) and the architecture of the neural network (Section 4.2) are presented. Following this, the research results are presented in Section 5, and several directions for future research are discussed in Section 6. The paper concludes by summarizing the key findings and insights drawn from the preceding sections—Section 7.

## 2. Digital Twin, Systems Medicine and Predictive Medicine

*2.1. Digital Twin, Systems Medicine and Predictive Medicine in Literature*

- Digital Twin

Regarding the application of Digital Twins in medicine, one of the central characteristics that has played a significant role in shaping this study has been specificity. Unlike other fields of application such as manufacturing, energy production, and smart cities, the specificity of Digital Twins in medicine leads to obtaining a complex image of the human body starting from genes, cells, organs, and organ systems, thus providing a personalized, patient-centered approach [9]. These data will be transformed into information and used for modeling and simulating patient behavior, continuously monitoring health, identifying pathologies, and providing personalized treatments. In specialized literature, various definitions for the human Digital Twin can be identified.

- Son et al. [10]—"A human Digital Twin could show what is happening inside the linked physical twin's body, making it easier to predict the occurrence of an illness by analyzing the real twin's personal history and the current context such as location, time, and activity".
- Ala-Laurinaho et al. [11]—"An emerging approach for disease treatment and prevention encompassing the use of new diagnostics and therapeutics, targeted to the needs of a patient based on their own genetic, biomarker, phenotypic, physical, or psychosocial characteristics".
- Bruynseels et al. [12]—"Human Digital Twins—the assumption that one is in possession of a data magnifying glass, that gives a detailed account of the molecular, phenotypic, and lifestyle history of persons".
- Shengli [13]—"A human Digital Twin is a copy or counterpart in cyberspace of a real person in our physical world. It is the digital description of you in a digital manner on a computer or server in the cloud. The model analyzes the timely data, historic data, the data from your relatives, and obtains insight from these data by Cloud Computing, Deep Learning, etc.".

According to the definitions presented earlier, creating a Digital Twin associated with the patient involves integrating various sources of data (biological data, genetic data, biomarkers, phenotypic characteristics, psychosocial data) and emerging technologies (Deep Learning and Cloud Computing) to obtain a holistic image of the human body. Thus, the Digital Twin is not just a virtual representation of the human body but serves as a dynamic reflection of the patient's health status and the evolution of their pathologies in real time. Continuous data monitoring aims to support clinicians in identifying risk factors and trends that lead to early diagnosis of diseases and provides various suggestions for personalized treatment. The Digital Twin can help reduce the risks associated with the

occurrence of side effects, improve treatment effectiveness, and offer new opportunities in terms of diagnosis, personalized treatment, and disease prevention.

- Systems Medicine

Systems Medicine [14] represents a systemic approach to modern medicine that integrates information from several sources such as systems biology and bioinformatics (genomic, transcriptomic, proteomic, metabolomic, and imaging data), as well as mathematical modeling at the physiological level for clinical applications. Using these integrative approaches, the functional and morphological structures of the organs can be analyzed, starting with DNA sequencing. Additionally, genetic pathologies that may appear at the level of these organs are detected. In the specialized literature, different definitions of Systems Medicine can be identified.

- Benson [15]—"Systems medicine encompasses the application of theoretical strategies through iterative and reciprocal exchanges of inputs between physicians, biologists, pharmacologists, bioinformaticians, and mathematicians in the fields of medical concepts, study, and practice".
- Cascante et al. [16]—"The driving force behind the eventual improvement of patient outcomes is, therefore, the recurrent interaction between bedside examinations, experimental models, and statistical analyses".
- Bousquet et al. [17]—"Systems Medicine is the latest definition of pediatric allergic diseases over a systematic translational approach involving detection, diagnosis, prevention, and therapy".
- Mayer et al. [18]—"System-based modeling also resulted from heterogeneous and daunting conditions, such as irritable bowel disease".
- Hood and Flores [19]—"Systems Medicine is a systemic approach to medicine and health that identifies all the components of a system and the interactions between them. Thus, the complex processes of the human body are characterized by interactions at the level of structural and functional organization".

According to the definitions presented earlier, Systems Medicine represents a medical paradigm that integrates knowledge from a diverse range of fields and promotes collaboration among physicians, biologists, pharmacologists, bioinformaticians, and mathematicians. By integrating laboratory research, clinical observations, and statistical analyses, Systems Medicine focuses not only on diagnosing pathologies and providing personalized treatment but also on prevention. The application of systemic models allows for the analysis of interactions between different levels of the organism, from the molecular and cellular levels to tissues and organs, and the identification of underlying connections. In conclusion, Systems Medicine can contribute to shaping a comprehensive perspective on the diagnosis, treatment, and prevention of diseases, laying the foundation for a patient-centered healthcare system.

Predictive Medicine is an approach to modern medicine that utilizes information about a person's genes, proteins, or clinical information to prevent, diagnose, or treat a disease. In [20], Valet and Tárnok highlighted that the goal of Predictive Medicine involves identifying shifts in a patient's health condition before any visible deterioration or improvement occurs in their current status. The authors presented the role of Predictive Medicine for patients of different ages (newborns, children, and the elderly) in the detection of complex diseases such as cancer, leukemia, diabetes, and asthma through their early identification and the adoption of therapeutic measures before the appearance of symptoms.

Thus, Digital Twin, Systems Medicine, and Predictive Medicine—Figure 1, redefine the paradigm of modern medicine, shaping an interdisciplinary and personalized approach to human health. The Digital Twin, through the integration and processing of data from patients, along with the technologies used, constructs a holistic picture of the patient's health status, reflecting the dynamism and interactions at the level of various components in real time. With the help of Systems Medicine, connections between the various levels of the human body are identified, highlighting the establishment of dependencies and rules

that lead to the prevention, and diagnosis of pathologies, and the patient's response to personalized treatment. Through the models and results provided by Predictive Medicine, personalized recommendations for diagnosis and treatment are offered, and by analyzing risk factors, predispositions to certain pathologies can be identified.

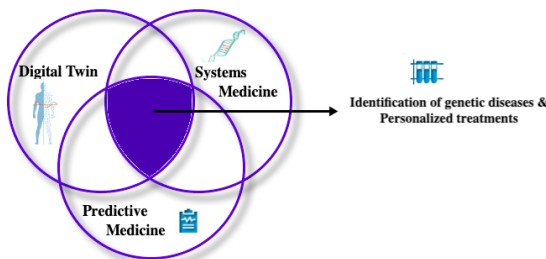

**Figure 1.** Interconnection of the Digital Twin, Systems Medicine and Predictive Medicine.

*2.2. Digital Twin, Systems Medicine and Predictive Medicine in Ophthalmology*

Following research on the application of the Digital Twin in medicine, we observed that one of the promising yet underexplored research directions is represented by ophthalmology, an area within Systems Medicine and Predictive Medicine where the utilization of Digital Twins holds particular promise. Starting from DNA sequencing, we can identify the gene mutations that predispose individuals specifically to glaucoma.

Glaucoma [21] is a progressive eye disease that leads to irreversible vision loss due to damage to optic nerve fibers. Risk factors contributing to the development of glaucoma include increased intraocular pressure, ocular trauma, chronic conditions like diabetes, prolonged corticosteroid treatment, myopia, hyperopia, and genetic predisposition. Various types of glaucoma have been previously identified.

- *Congenital glaucoma* is a rare eye condition (optic neuropathy) that predominantly affects males. This disease may manifest during the neonatal or early infantile period and is characterized by symptoms such as sensitivity to light and ocular discomfort. *Primary congenital glaucoma (PCG)* is an autosomal recessive disorder largely attributed to mutations in CYP1B1, and to a lesser extent, in LTBP2, TEK, MYOC, and FOXC1 [22].
- *Primary open-angle glaucoma (POAG)* [23] is the most common form of glaucoma and is associated with increased intraocular pressure (IOP) (>21 mmHg) and an open iridocorneal angle (35°–45°). Another manifestation is central retinal venous occlusion (the stoppage of circulation in the central vein of the retina or in one of its branches); heredity is one of the factors that favor the appearance of this pathology. In [24], Sears et al. highlighted that mutations in each of the three genes, myocilin (MYOC), optineurin (OPTN), and TANK binding kinase 1 (TBK1), can cause primary open-angle glaucoma (POAG).
- *Primary closed-angle glaucoma (PCAG)* is determined by an anatomical predisposition of the eyeball, which leads to partial or total obstruction of the drainage of the aqueous humor by blocking the angle at the periphery of the iris.
- *Secondary open-angle glaucoma* can be associated with intraocular tumors and eye hemorrhages, or it can be caused by treatment with cortisone and post-laser procedures.
- *Secondary closed-angle glaucoma* can be caused by pupillary block, which is determined by intumescent cataract, or it can occur without pupillary block, as in neovascular post-inflammatory glaucoma.

Thus, through the analysis of genomic data and multimodal data, glaucoma can be predicted, prevented, identified, and managed. Moreover, personalized treatment plans based on genetic characteristics should be developed to identify the most effective treatment solutions. Continuous monitoring of the evolution of the pathology is achieved by analyzing real-time data received from the patient which are integrated with their medical history, as well as data obtained from regular medical examinations.

As stated in the Section 1, there are no sensors capable of acquiring real-time information directly from the eye, such as intraocular pressure, cup/disc ratio, or iridocorneal angle, so monitoring the progression of glaucoma will be exclusively conducted during medical consultations by the ophthalmologist. Thus, the absence of sensors for real-time monitoring represents a limitation in managing this pathology. The implementation of a Digital Twin focused on diagnosing glaucoma will utilize an integrated approach, conducted with continuous communication with medical personnel who will assess relevant parameters and adjust treatment according to the progression of the disease. The ophthalmologist will provide continuous feedback to the Digital Twin system regarding treatment suggestions, which contributes to training the system so that through ongoing data collection and analysis processes, it improves the accuracy of the results provided.

Regarding the use of the Digital Twin for diagnosing glaucoma, this could be applied both in the ophthalmologist's office and on a large scale in medical clinics or hospitals.

The use of the Digital Twin in an ophthalmologist's office emphasizes the importance of direct interaction between the doctor and the patient, facilitated by the doctor's continuous access to the patient's medical history and immediate interpretation of the results provided by the Digital Twin. However, one limitation is that the Digital Twin system can only be trained with data obtained from the respective doctor's patients. To provide results with a high degree of accuracy, the system needs to be trained with large volumes of data, suggesting that the use of Digital Twins for diagnosing glaucoma would be more efficient in medical clinics and hospitals. In this case, within a large number of patients, the Digital Twin could be used to automate the diagnostic process and reduce the time spent in ophthalmologic consultation, while the final decision still remains solely with the specialist doctor.

Regarding the costs of using a Digital Twin in an ophthalmologist's office, one must consider the initial expenses related to acquiring equipment for collecting patient data, specialized software for processing and analyzing data, as well as for developing and implementing prediction and diagnostic algorithms. Other costs could include training qualified personnel who will utilize the data provided by the Digital Twin, thereby incurring maintenance costs. At the clinic level, costs will be much higher as the system needs to serve a larger number of doctors and patients, simultaneously managing large volumes of data. Thus, the costs of acquiring the infrastructure capable of supporting a large number of users increase significantly. In both cases, to avoid ethical issues, a budget must be allocated for implementing security measures, including data encryption and confidentiality, access monitoring, security testing, and cyber attack detection.

Thus, the Digital Twin represents a decision support tool for physicians, providing them with real-time predictive analyses regarding diagnosis and personalized treatment, facilitating the communication between the doctor and patient both in the physician's office and in medical clinics.

## 3. Architectural Framework for Digital Twin, Risks and Vulnerabilities

### 3.1. An Architectural Framework for Digital Twin in Ophthalmology

The architecture proposed (Figure 2) in this chapter is based on the architecture presented in [25] and facilitates the construction of a Digital Twin associated with the patient, aimed at providing decision support to the physician in diagnosing glaucoma. From a structural point of view, it is composed of five layers: Data Acquisition and Dissemination, Data Management and Synchronization, Realization of the Digital Twin Associated with the Patient, Virtualization and Accessibility, and a Comprehensive Layer for Enhanced Security and Cybersecurity, which provides a systematic view of the system. The initial layers focus on collecting and managing patient data, while in subsequent layers, these data will be used to construct a Digital Twin associated with the patient, integrating various parameters for glaucoma diagnosis. The final layer of the architecture focuses on the security of the entire system, aiming to ensure users' data confidentiality, integrity, and availability. All these layers provide a robust framework for constructing the Digital Twin.

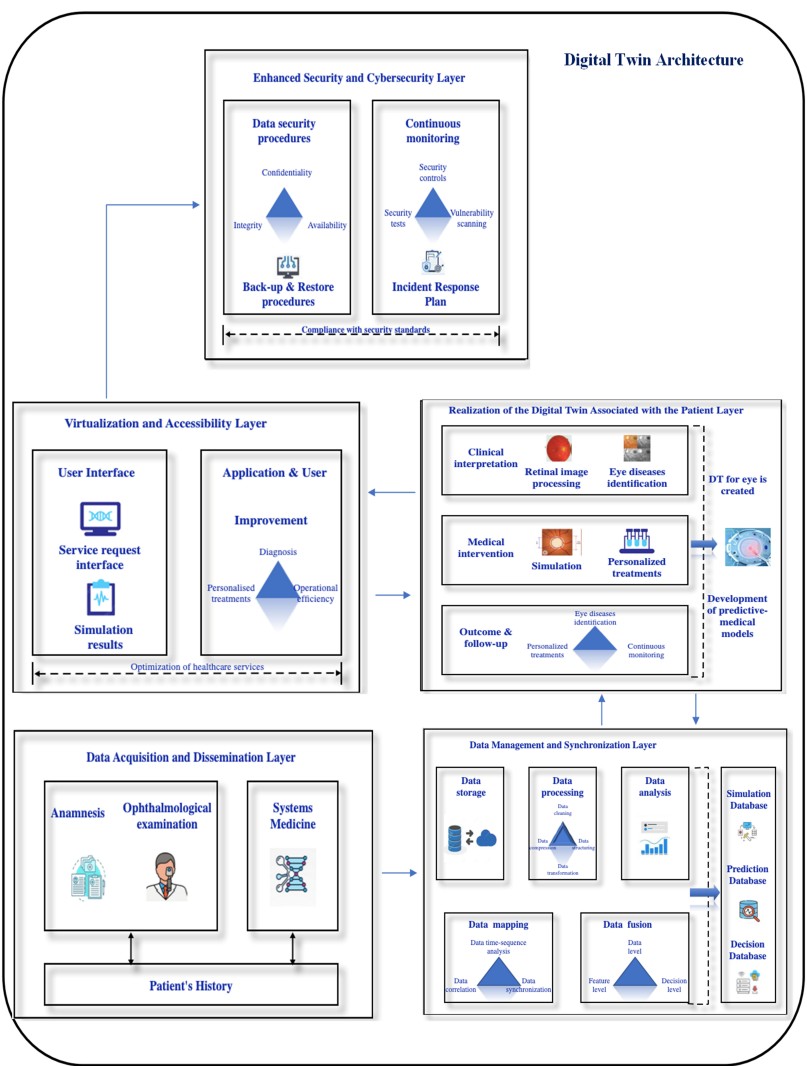

**Figure 2.** Digital Twin Architecture.

The first layer is represented by *Data Acquisition and Dissemination*—Figure 3. At this stage, the emphasis is on the patient's medical history, initial evaluation, data related to previous eye disorders, current medications, as well as corneal and pupil dimensions. These data will be used for constructing a Digital Twin associated with the eye.

The second stage involves acquiring data from the ophthalmological examination—Tabel 1, which includes measuring intraocular pressure and testing visual acuity. If the eye's anatomy suggests a predisposition to primary open-angle glaucoma, Pentacam is used. The irido-corneal angle (whether closed or open) is also evaluated, followed by a FO (funduscopic examination) to determine the cup/disc ratio (c/d) implicitly detecting the presence of optic disc hemorrhages. The c/d ratio typically ranges between 0.3 and 0.4 for normal values, but for patients with glaucoma, it increases, often reaching 0.6 to 0.8.

Corneal pachymetry will be conducted to measure the thickness of the cornea, as a thin cornea (below 550 microns) is a risk factor for the progression of open-angle glaucoma. Additionally, OCT (Optical Coherence Tomography) of the optic nerve or retina will be performed to detect structural changes that may occur at the level of the optic nerve fibers. If the patient is in the follow-up stage of this pathology, this information will be added to the patient's medical history.

By employing Systems Medicine, the model of the Digital Twin associated with the eye will incorporate the patient's genetic information, including the presence of gene mutations that predispose to glaucoma. Consequently, following the integration of genetic factors, the

focus shifts to a systemic approach that forms the basis for predictive medicine. It aims to achieve the early diagnosis of various types of glaucoma, personalized treatment provision, and monitoring of the patient's response to prescribed medication.

**Table 1.** The patient's acquired data for glaucoma identification.

| Stage | The Patient's Acquired Data |
|---|---|
| Information about the patient | • Patient's medical history<br>• Data related to previous eye disorders<br>• Current medications |
| Data acquisition from the ophthalmological examination | • Measurement of intraocular pressure<br>• Testing visual acuity<br>• Evaluation of irido-corneal angle<br>• Determining cup-disc ratio<br>• Measurement of corneal thickness<br>• Detection of structural changes at the level of the optic nerve fibers |
| Genetic Information | • Presence of gene mutations predisposing to glaucoma |

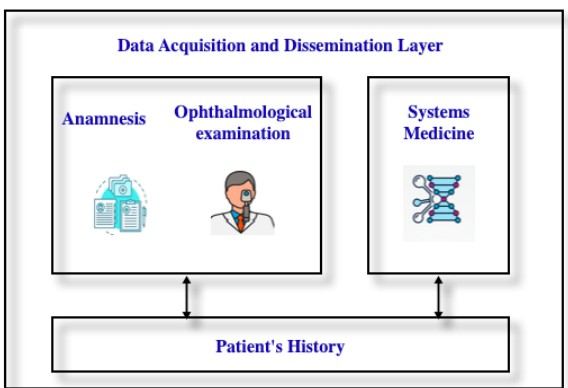

**Figure 3.** Data acquisition and Dissemination Layer.

Once these data are acquired, they are transmitted to the *Data Management and Synchronization Layer*—Figure 4, which incorporates functionalities such as data storage, processing, mapping, and fusion. Simultaneously, the security, authenticity, and integrity of the data are ensured using encrypted communication protocols, at least TLS 1.1 (Transport Layer Security). Additionally, at the database level, it is recommended to implement backup and restoration procedures. The implementation of backup and restore procedures contributes to maintaining operational continuity in exceptional situations arising from unexpected incidents such as cyber-attacks or temporary data unavailability. All these events lead to significant delays in the glaucoma identification process and to the blocking of access to information. Also, from a legislative perspective, the implementation of backup and restore procedures is carried out in accordance with medical security standards, aiming to reassure patients that their sensitive information is adequately protected and to support the quality of medical services.

Recurrent testing and updating of the software will be performed through Penetration Testing and periodic vulnerability scanning. The data from the patient's medical history and ophthalmological examination are processed and analyzed. Subsequently, they are utilized in the simulation, evaluation, and prediction of potential eye pathologies. By conducting these security tests, we identify potential vulnerabilities that attackers could exploit, which is essential for preventing unauthorized access to sensitive patient data. Identifying and addressing vulnerabilities, as well as retesting the system, ensure users

have a high level of security regarding the use of the Digital Twin. The data obtained from medical consultations represent sensitive information for glaucoma identification, and any modification or unauthorized access to these data can affect the integrity and reliability of the Digital Twin.

Once storage in a database is complete, the data are preprocessed to eliminate redundant information. Data cleaning methods can be employed, along with data compression, smoothing, reduction, and transformation techniques. One of the reasons that justifies performing the data cleaning process is to ensure their quality. Data cleaning involves removing deficiencies and inconsistencies that arise during the data collection process. Data standardization is carried out to comply with the required formats that will later be incorporated into the prediction algorithms used by the Digital Twin. Another step in the data cleaning process involves reducing their size and complexity. Removing irrelevant information and applying compression and optimization techniques leads to improving the accuracy and performance of the Digital Twin.

Subsequently, the data undergo analysis using various statistical methods (such as correlation analysis, regression analysis, and discriminant analysis) or neural network approaches (including neural networks based on gradient algorithms and optimal regularization methods like feedback networks such as the Hamming network and wavelet neural network) [26]. Furthermore, data analysis using the statistical methods mentioned earlier is carried out to detect incomplete data stored in the database and to identify discrepancies between the datasets used. Additionally, this analysis contributes to ensuring high data accuracy, which will influence the efficiency of prediction algorithms for glaucoma detection. Another argument supporting the data analysis process using statistical methods is the identification of correlations and relationships between parameters to contribute to identifying trends and patterns associated with glaucoma diagnosis.

Data mapping involves synchronously mapping and correlating physical data with virtual operations, based on data storage and processing. It encompasses data time sequence analysis, data correlation, and data synchronization [27].

The methods used for data fusion include synthesis, filtering, correlation, and integration. Data fusion can be performed at both the raw data level and the decision level using various methods such as Kalman filtering, Bayesian estimation, classical reasoning, and artificial intelligence. Working with real-time data may pose storage-related problems, while complex processing algorithms can lead to issues related to massive data processing [26].

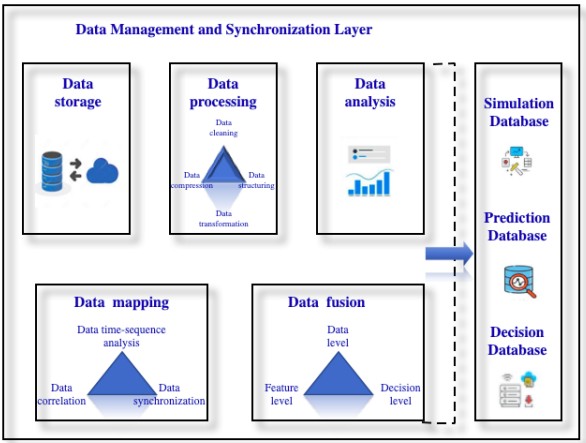

**Figure 4.** Data Management and Synchronization Layer.

As a result of the data processing carried out in the second layer, predictive medical models will be developed in the *Realization of the Digital Twin Associated with the Patient Layer*—Figure 5. These models can highlight structural and functional changes that may occur at the eye level, aiding in the continuous and predictive monitoring of the patient's health status. The Digital Twin will send alerts to the medical staff to illustrate the predicted

behavior of the patient regarding the progression of the pathology or the adjustment of treatments depending on the context.

At this layer, patient data have already been processed, and the use of Machine Learning assists in developing personalized predictive models for each patient. These predictive models are trained to recognize glaucoma based on the patient-derived data. Additionally, the use of Machine Learning for integrating glaucoma prediction contributes to the development of personalized decision support systems based on various demographic parameters such as age, gender, race, as well as other medical conditions associated with the goal of providing individualized treatment and management recommendations.

Retinal images will undergo processing, utilizing the insights presented in [28] as a foundational reference. From these images, anatomical features such as Optic Cup to Disc Ratio (CDR), Retinal Nerve Fiber Layer (RNFL), Peripapillary Atrophy (PPA), Neuroretinal Rim Notching, and Vasculature Shift will be automatically extracted to diagnose potential cases of glaucoma. Various methods are employed for optic disc detection and center localization, including Local Contrast Color Enhancement, Thresholding Highest Pixel Intensities, and Shade Correction using morphology. As detailed in [28], the highest accuracy is achieved using the latter method, with an impressive percentage of 94.7%. For PPA detection, Haleem et al. [28] employed the Disc Difference Method, yielding an accuracy of 95%, while GLCM-Based Texture analysis resulted in 92.5% accuracy. Additionally, GLCM-Based Texture analysis was utilized for PPA extraction, achieving an accuracy of 73%. The employed extraction methods encompass Directional Gabor Filters, Markov Random Fields, and Intensity Profile Plotting, with the latter method attaining an accuracy of 91.5%.

In addition, at this layer, predictions will be made to illustrate how Primary Open-Angle Glaucoma evolves over the next five years if the patient complies with the recommended medical treatment [29]. To make these predictions, linear regression was used.

To identify the evolution rules of the patient's data, a mining algorithm with sequence mode was used. Once the Digital Twin associated with the eye is created, the data generated during periodic eye examinations can be used to evolve the model. By tracking the glaucomatous status through IOP measurements and data from medical imaging, both the patient and the doctor will receive information about the recommended medication or surgical intervention, as well as the interval until the next consultation [30].

- If the IOP has not increased compared to the previous examination conducted no more than six months ago, the patient will be advised to undergo another ophthalmological examination after six months.
- If the IOP has not increased compared to the previous examination conducted more than six months ago, the patient will be advised to undergo an eye examination after 12 months.
- If the IOP has increased compared to the previous examination, the patient will be advised to undergo an eye exam after 1–2 months.
- If the IOP has not increased compared to the previous examination, but there has been progression in other aspects, an ophthalmological examination will be recommended after 1–2 months.
- If the IOP has not increased compared to the previous examination and there has been no progression in other aspects, an ophthalmological examination will be recommended after 3–6 months.

In addition, the most appropriate treatment options will be suggested to the patient, including local medication (administering daily drops every 8, 12, or 24 h throughout life), general medication (tablets or intravenous infusions to lower intraocular pressure), laser therapy, or surgery.

The fourth layer in the Digital Twin architecture, the *Virtualization and Accessibility Layer*—Figure 6, associated with the eye, contributes to improving operational efficiency and facilitates the identification, diagnosis, and provision of personalized treatments to patients.

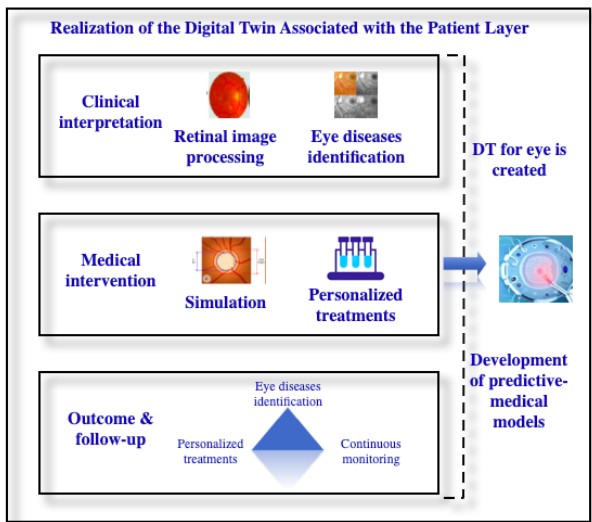

**Figure 5.** Realization of the Digital Twin Associated with the Patient Layer.

This layer contains two blocks: *User Interface* and *Application and User*. The User Interface contains a service request interface through which both medical staff and patients can follow the updates that have been realized. In addition, a service and platform operation interface is provided, where stakeholders can follow the method of glaucoma diagnosis and treatment suggestions. The application and the user provide user access to interact with different CloudDTH platforms while supporting efficient task coordination [31]. Hence, end users, entities, and processes are provided with access to observe simulation results from digital models, enabling them to make decisions concerning the monitoring of patient pathologies. By employing diverse methods of data visualization and accessibility, the Digital Twin facilitates the optimization of healthcare services.

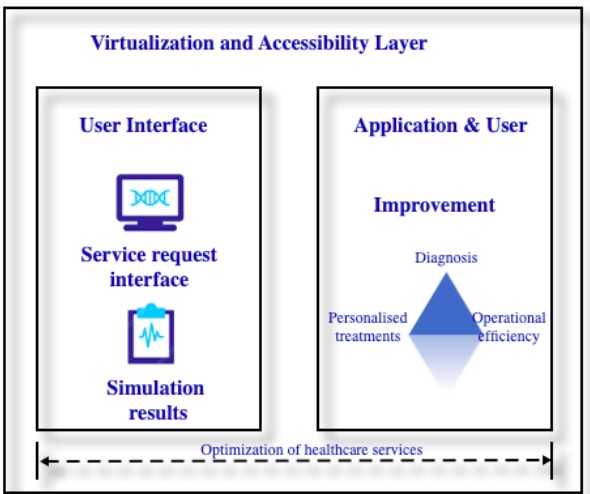

**Figure 6.** Virtualization and Accessibility Layer.

The layer for *Enhanced Security and Cybersecurity*—Figure 7, represents the last layer in the Digital Twin architecture. It focuses on ensuring the security of the entire system, platform, and network, including the user data. Furthermore, it seeks compliance with security standards while ensuring confidentiality, integrity, and availability of data. Data security procedures are implemented at the level of this layer in terms of user authentication, data encryption, monitoring, and auditing, implicitly managing their lifecycle. In addition, security controls are implemented to ensure protection against cyber-attacks. Periodic

security testing is recommended to identify potential vulnerabilities at each layer of the Digital Twin.

Recurring backup and restoration procedures should be carried out to avoid the problems that may arise as a result of data theft, modification, or deletion. At the management level, the aim is to implement an incident response plan that contains the presentation of the way to respond in the event of cyberattacks, respectively, non-compliance with data security regulations such as GDPR (General Data Protection Legislation), HIPAA (Health Insurance Portability and Accountability Act), or PCI DSS (Payment Card Industry Data Security Standard). Also, in the medical field there are standards for the exchange of information such as HL7 (Health Level Seven International), and IHE (Integrating the Healthcare Enterprise), respectively, "standardized management of the system" to standardize the way of collecting, sharing and managing medical data.

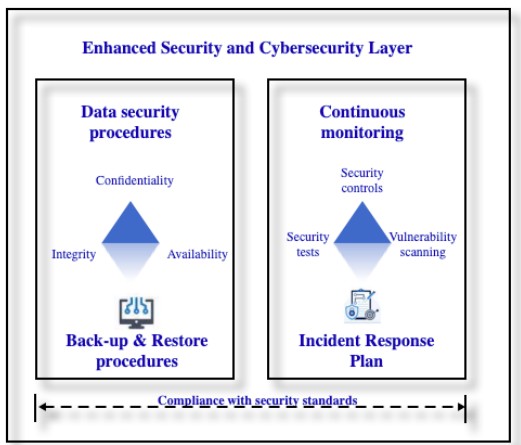

**Figure 7.** Enhanced Security and Cybersecurity Layer.

*3.2. Risks and Vulnerabilities at the Digital Twin Level*

- The risks of doctor reputation and patient health in Digital Twin for glaucoma detection

Using the Digital Twin for detecting glaucoma and providing treatment suggestions poses a series of risks, which will be outlined below. As mentioned earlier, the Digital Twin serves as a decision support tool for physicians, tasked with conducting predictive analyses regarding glaucoma detection and providing various scenarios for its progression. Thus, the use of a Digital Twin in this context cannot directly influence the patient's health status. However, the decision-making process may be affected by errors occurring during the data collection process from the patient, which will train the algorithm responsible for detecting glaucoma and providing treatment suggestions, thus impacting the relationship between the physician and the patient.

One of the risks could be the constant provision of inaccurate information by the Digital Twin. Emphasizing that the physician is the one proposing the optimal treatment solution, the patient might perceive the version suggested by the Digital Twin as favorable. This can lead to a decrease in the patient's trust in the physician's ability to provide a correct diagnosis and prescribe appropriate treatment.

Initially, the Digital Twin system will provide diagnostic and treatment information with a low level of accuracy. With the continuous training of the system using feedback from the specialist physician, the accuracy level of the system will increase. Thus, processing large amounts of data from both patients and medical personnel may cause delays in the diagnostic and treatment procedure provided by the Digital Twin. In this context, the doctor may wait for the response provided by the Digital Twin, which could lead to excessive dependence on technology. From the patient's perspective, trust in the specialist doctor may be diminished, resulting in a decrease in reputation.

Another risk could be represented by the confidentiality of the data obtained, stored, and processed during the ophthalmologic consultation by the Digital Twin in accordance with legislative standards. In this case, the risk of damaging the doctor's reputation could be represented by the lack of trust from the patient in the medical staff's ability to protect sensitive information and use it for personal purposes.

Also, one of the paramount elements in the relationship between doctor and patient is transparency and communication. Therefore, the patient desires the doctor to consistently explain how the Digital Twin provides the diagnosis and offers treatment suggestions. The lack of effective communication and discussion based on the suggestions received from the Digital Twin can lead to the formation of an opinion where the doctor is perceived as lacking professionalism and not providing the best treatment solution.

In conclusion, to avoid the occurrence of risks that could diminish the reputation of the specialist doctor, the emphasis is on finding a balance between the use of technology and the clinical skills of the doctor. This involves making decisions ethically and responsibly, alongside maintaining genuine communication with the patient.

- Cybersecurity challenges and research frontiers for Digital Twin

The identification of risks is carried out by taking into account all the information contained in the assets, parties involved, and contractors, but also as a result of the exploitation of threats and existing vulnerabilities. The sources of the risks will be identified (human errors, component failures), the consequences, the results, the impact (unavailability of services, loss of data, theft of information), the reasons that led to their appearance (incorrect application of the prediction algorithm, training prediction algorithm with modified data), protection mechanisms, existing controls (control systems, detection, compliance with procedures, standards), and the time and level of the Digital Twin at which they can occur [32].

Threats to physical assets, service unavailability, data compromise, technical failures, internal threats (usually caused by insiders leading to information extraction), and external threats (hardware component failures, software errors, asset deletion, criminal acts, and espionage) [33] can impact the cybersecurity of a digital twin system.

Human Digital Twins are critical systems where confidentiality (C), integrity (I), and availability of data (A), as well as entity assets (E) and their location (L) are emphasized. In addition, security threats affect the operational requirements of Digital Twins, such as their performance, reliability, maintainability, and interoperability. It is recommended to perform security analysis at the Digital Twin layer level. Large-scale attacks mainly target patient data processing, predictive medical models, multimodal retinal image processing algorithms, artificial intelligence, virtualization platforms, and networks. The attack can be launched at the level of the physical asset (at the equipment level where patient data will be received as a result of the medical examination) and its digital counterpart. At the Digital Twin level, there can be exploits for all the components that provide resources for distributed and centralized computing for the entire network, as well as for security settings that can enable the execution and management of critical data involved in various decision processes. Regarding the compromise of the Digital Twin, one can consider the existence of an attack at the level of the physical asset by modifying the input data carried out in layer 1 implicitly in the processes of processing and representing them in layers 2–4. Moreover, the existence of an attack on its digital counterpart that will occur in layers 2–4 will modify the outputs that will then be transmitted to layer 1, thereby not achieving the desired behavior of the physical asset. Thus, Digital Twin cannot assist medical staff in diagnosing and prescribing personalized treatments. In [33], Alcaraz and Lopez identified a series of vulnerabilities that can appear at the Digital Twin level, which are highlighted in Figure 8.

Periodic vulnerability scanning at the level of each layer of the Digital Twin is recommended to identify the misconfigurations and security issues of physical and virtual assets. Once the potential vulnerabilities have been identified, the focus is on fixing them by implementing corrective measures, such as software updates, appropriate system configuration,

and default user access policy. The continuous monitoring and reporting of vulnerabilities are important steps in protecting against cyberattacks. Another way to identify threats through exploitation is to become the source of cyberattacks, represented by security tests. The first stage is the identification of potential threats to which the Digital Twin is subjected, which contributes to the development of scenarios and the choice of testing methodology. Depending on the type of test chosen, which can take place at the level of the Digital Twin infrastructure, the network, the construction of medical-predictive models, the storage platform, and the data visualization, the cyberattack will be simulated. Following the simulations, potential threats are identified, and a report containing the risks, vulnerabilities, and recommendations for their remediation is produced. After the implementation of the suggested measures, retesting and continuous monitoring are recommended.

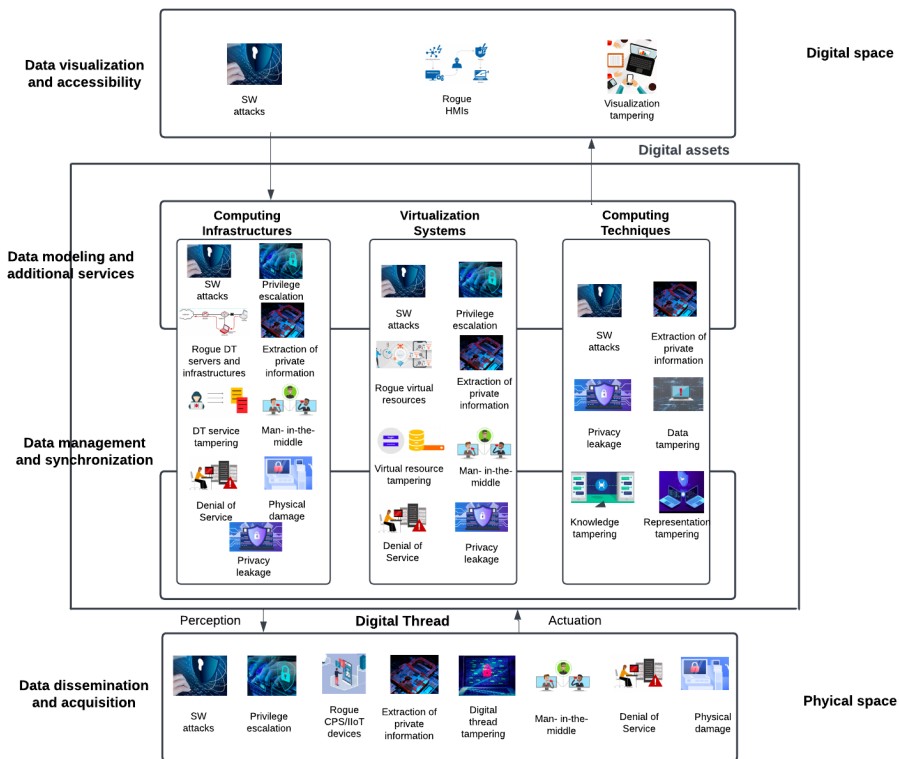

**Figure 8.** Vulnerabilities at the Digital Twin level.

Thus, through periodic vulnerability scanning at the Digital Twin level, both the security, integrity, and confidentiality of patient data, as well as medical processes, can be ensured. To improve security measures, a multi-level approach is proposed, involving both implementing security controls at each layer of the Digital Twin and taking corrective actions and continuous monitoring of the system. As a result of identifying vulnerabilities, the corrective measures that need to be implemented are those mentioned earlier, referring to software updates, system configuration adjustments, and the implementation, and implicit compliance with access control policies. Continuous monitoring of the system contributes to the real-time detection of suspicious activities and allows timely remedial actions to be taken. Also, following the security tests, the generated reports contain information about the risks and vulnerabilities identified, providing recommendations for addressing these situations.

## 4. Materials and Methods

In the context of this study, the focus was on outlining an architectural framework for creating a Digital Twin associated with the patient, integrating concepts of Predictive

Medicine. Thus, a methodological approach was conducted, starting with the selection of datasets used for conducting statistical analyses based on various demographic parameters such as age, gender, and race of the patients. The study continues with the training of the neural network aimed at identifying the presence of glaucoma and presenting the obtained results.

### 4.1. Datasets

The dataset Harvard Glaucoma Fairness with 3300 Samples (Harvard-GF3300) [34,35] was used to perform the statistics. The data categories contained in this set are represented by OCT (Optical Coherence Tomography) RNFLT map (Noncontact and Noninvasive Retinal Nerve Fiber Layer) of size 200 × 200, glaucomatous status, mean deviation value of visual field, 52 total deviation values of visual field, patient age, male, patient race, marital status, ethnicity, and language. Among the previously presented categories, glaucomatous status, patient age, male, patient race, and marital status were selected for the statistics. The analysis of the results is presented in Section 5.1. The dataset Harvard Glaucoma Detection with 500 Samples (Harvard-GD500) [36,37] was used to train the neural network. The dataset composition is presented in Section 4.2, while the results are presented in Section 5.2.

### 4.2. Convolutional Neural Network Architecture

We will describe the neural network used for predicting the presence of glaucoma. Our model was trained on the Harvard Glaucoma Detection (Harvard-GD500) dataset [36,37], which contains 500 samples from 500 patients for glaucoma detection. The dataset samples consist of retinal nerve fiber layer (RNFL) grayscale maps, each having a dimension of 225 × 225 pixels, accompanied by their corresponding visual field mean deviation and a label indicating whether the patient has been diagnosed with glaucoma or not.

Because the dataset contains mixed data, including both images and numeric values corresponding to the visual field mean deviation, we opted to use a multiple-input convolutional neural network (CNN). Our initial step involved data normalization for consistency. The RNFL grayscale image maps were normalized to the [0, 1] interval, while the visual field MD values were normalized using the L2 norm to the [−1, 1] interval.

After normalizing the dataset, we split it into subsets as follows:

- Training—80% of the dataset was used to train the network.
- Validation—10% of the dataset was used for fine-tuning the model's hyperparameters and preventing overfitting during the training phase. This helps assess the model's performance on unseen data during training. Unfortunately, the dataset only contains 500 samples, so we compromised and used only 10% for validation to avoid underfitting.
- Testing—10% of the dataset was used to test the model. This serves as an independent dataset, not used during training and validation, and is exclusively used after model training to evaluate its performance on entirely new, unseen data.

The network architecture can be seen in Figure 9. We used the functional approach in TensorFlow to create a multiple-input network.

The first branch of the network is dedicated to processing the RNFL maps. Convolutional layers are used to extract features related to the hierarchical representation and spatial hierarchy of the data. Early layers tend to learn simple features such as edges or corners, while deeper layers capture more complex and abstract features by combining information from previous layers. Convolutional layers maintain the spatial relationship between pixels. By using small filters, they capture local information while preserving the spatial arrangement of features, which is crucial for understanding visual patterns. Compared to fully connected layers, convolutional layers reduce the number of parameters by sharing weights across the input space, making them computationally efficient and capable of learning from large image datasets without excessive demands on memory. The stride dimension used for all convolutional layers is (3, 3). After the convolution layers,

we use a flattening layer to prepare the data for the dense layers, followed by several fully connected (dense) layers.

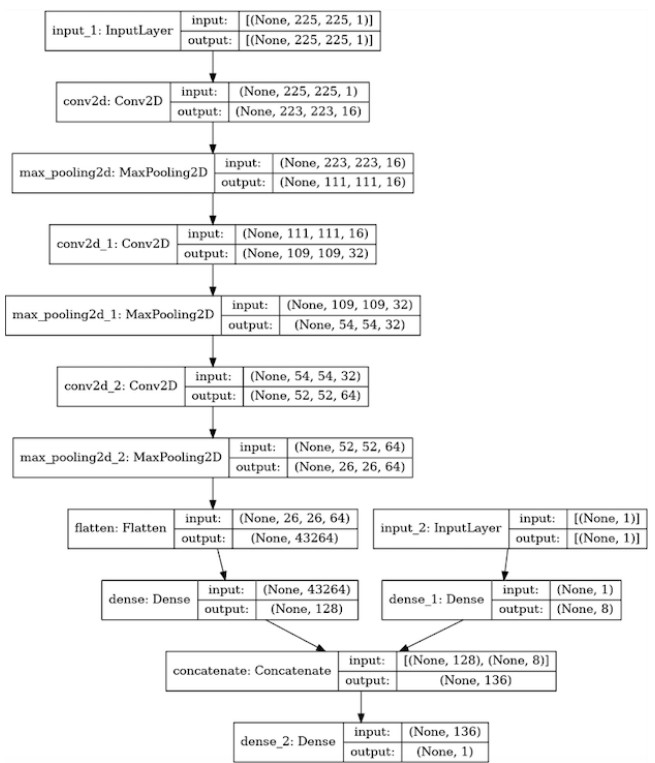

**Figure 9.** The network architecture.

The second branch of the network is dedicated to processing the visual field mean deviation values. This network is simple because we do not have too much data. It consists only of one fully connected layer.

The outputs of the branches are concatenated, and the combined output is fed to a fully connected layer with only one neuron, which serves as the output of the network. It uses the sigmoid activation function, commonly used for binary classification tasks. This function compresses the output of a neuron to a range between 0 and 1, effectively converting the network's output into a probability. The resulting value represents the probability of the patient having glaucoma.

We used the binary cross-entropy loss function, which is suitable for binary classification problems. It measures the difference between the predicted probability distribution and the actual distribution for a binary classification task. Specifically, it quantifies the distance between the predicted probability (computed by the model) and the actual target value for each example. This loss function works well when the model is predicting the probabilities for two classes (0 or 1). It encourages the model to learn to output probabilities that are as close as possible to the true labels for each sample.

## 5. Results

The overall objective of this study was to identify glaucoma and monitor its progression. However, one of the challenges is the application of new artificial intelligence techniques and building a trained model to predict one of the possible progressions based on various parameters such as IOP, corneal thickness, iridocorneal angle, C/D ratio, starting from the initial data. Considering the large volumes of ophthalmic data and the complexity of image processing for glaucoma diagnosis, we propose a divide et timpera approach, which involves finding subtypes centered on gender, age, race, and marital status, and training the network on these subtypes. Training the model based on these categories leads to the construction of a complex model. By using combined learning and prediction

algorithms, medical staff will be able to visualize patient status information, having the opportunity at any time to influence the prediction process by modifying relevant information. Otherwise, the Digital Twin does not replace the doctor, its role being to suggest the evolution of the pathology and how the personalized treatment offered influences this evolution. In other words, the Digital Twin serves only as a decision support element for medical staff.

In this chapter, the two stages that contribute to shaping the architectural framework are presented, essentially representing the results of the article. In the first section, an approach to data analysis and interpretation is presented as a result of the realization of statistics based on demographic parameters such as age, gender, marital status, and race. As a result of the obtained statistics, correlations have been identified that will contribute to the development of a Digital Twin, starting from the segmentation of patients into different categories.

The second section focuses on training a neural network based on 500 RNFLT maps and their corresponding visual field (MD), aiming to assist in the glaucoma identification process. Additionally, aspects related to the configuration and implementation of the network as well as the accuracy of the results obtained are presented.

### *5.1. Statistics on the Occurrence of Glaucoma Based on Demographic Factors*

The construction of a Digital Twin associated with the patient aimed at diagnosing glaucoma can begin with the analysis of statistics based on demographic parameters such as gender, age, and race. Based on the results obtained from these statistics, possible correlations between the mentioned parameters and the likelihood of developing this condition will be identified. All these contribute to forming a holistic view of the factors that favor the onset of the pathology and to identifying groups of patients predisposed to glaucoma.

As stated in Section 4, the dataset utilized for conducting the statistical analysis was the Harvard Glaucoma Fairness dataset with 3300 samples (Harvard-GF3300) [34,35], the details of which are also provided in this chapter.

From the data of the 3300 patients, 1748 patients suffering from glaucoma were identified, representing 52.97% of the total. Regarding the gender distribution among the patients diagnosed with glaucoma, 934 were women (53.43%) and 814 were men (46.57%). In terms of gender, uniformity in the incidence of glaucoma was thus identified, suggesting that there is no notable difference between these two demographic groups. The absence of a notable difference in the frequency of glaucoma cases between women and men can be attributed to common risk factors such as genetic predisposition and elevated intraocular pressure. Additionally, socio-economic factors such as education, income, and similar access to medical services may contribute to these outcomes.

Another factor that can be considered for conducting this study could be the age of the patients. The dataset used contains information about patients aged between 10 and 98 years old. Figure 10 depicts a distribution based on gender and the age category to which these individuals belong. Although the largest number of patients included in the study belong to the age category over 60 years old, it is still around this age that the highest trends of the increase in the number of glaucoma occurrences are detected.

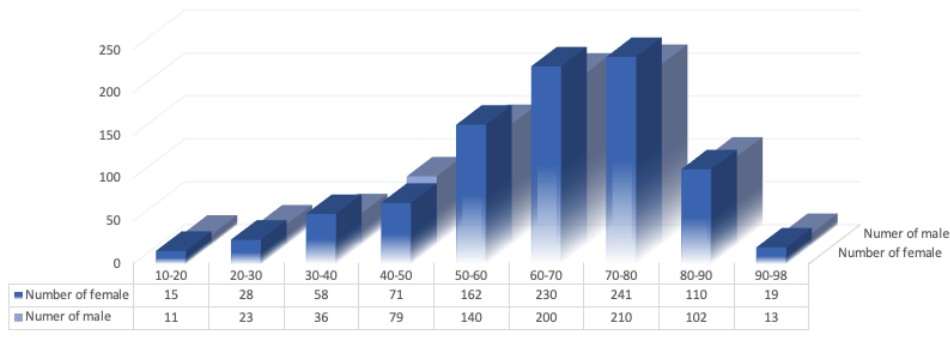

**Figure 10.** The occurrence of glaucoma based on gender and age category.

According to Figure 11, regarding the total number of glaucoma cases identified in the dataset, the highest percentage is noted for patients aged 70 to 80 years (26%), followed by the category of 60 to 70 years (25%). According to the data presented already in the "Introduction", we can observe a growing trend in the number of glaucoma cases after the age of 50, increasing significantly for individuals in the age groups of 60–70 and 70–80 years old. Thus, the age of 60 can be considered a transitional moment regarding the risk of glaucoma onset, primarily caused by the aging process affecting both the eye's structure and function, recommending regular ophthalmic check-ups at regular intervals. Another factor can be represented by the fact that in the early stages, this pathology is asymptomatic and can progress slowly. At the opposite pole, the fewest cases of glaucoma were identified among people aged between 10 and 20 years (1%) and those aged between 90 and 98 years (2%).

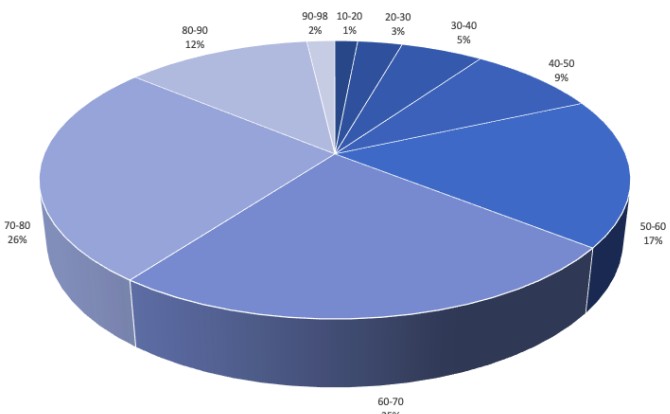

**Figure 11.** The occurrence of glaucoma based on age category.

Another demographic factor considered for this study is race, with the three races being White or Caucasian, Black or African American, and Asian. In the dataset used, the total number of individuals for each race is the same, namely 1100. According to the data from the dataset, 48.7% of individuals of White or Caucasian race involved in the study were diagnosed with glaucoma, 62% of those of Black or African American race, and 48.1% of those of Asian race. In the case of individuals diagnosed with glaucoma, a distribution based on race and gender is presented in Figure 12.

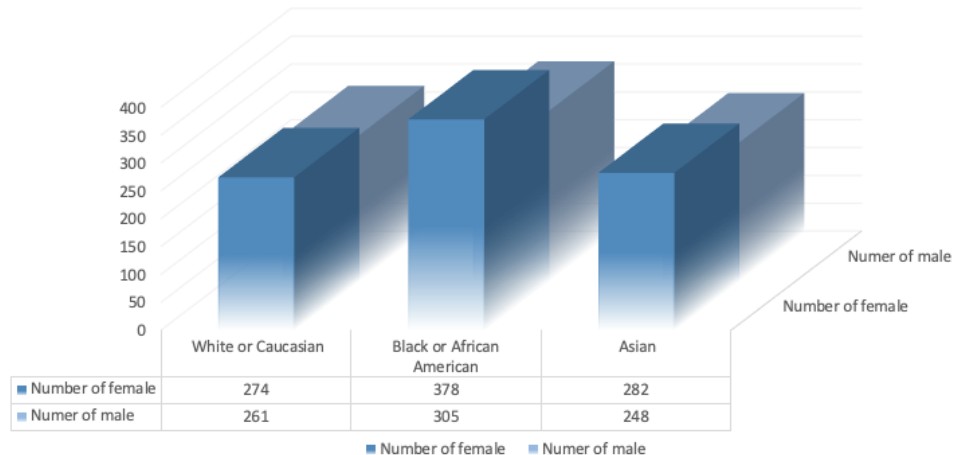

**Figure 12.** The occurrence of glaucoma based on gender and race category.

It is observed that there are variations in the percentage of glaucoma occurrence depending on race, with the highest number of cases among individuals of Black or African American race—39.07%, followed by individuals of White or Caucasian race—30.61%, and individuals of Asian race—30.32%, all these aspects being highlighted in Figure 13. Through an analysis considering the gender of the individuals involved, it can be noticed that in all three cases, the frequency of glaucoma occurrence in men is higher than the frequency of glaucoma occurrence in women. Thus, these variations may be caused by the presence of hereditary factors that predispose to the occurrence of glaucoma. Another factor could be represented by the structure of the eye, which in this situation presents a narrower drainage angle, leading to increased intraocular pressure and favoring the occurrence of glaucoma. According to the medical literature, Sample et al. [38] mention several differences between races regarding the onset and monitoring of glaucoma. In this category, differences have been identified concerning optic nerve appearance, visual field results, and other clinical signs and risk factors. Thus, during the ophthalmologic consultation, larger refractive errors, an increased cup-disc ratio, and the presence of a thin cornea can be identified. However, socio-economic factors must also be taken into account, as they play a crucial role in terms of population access to medical services.

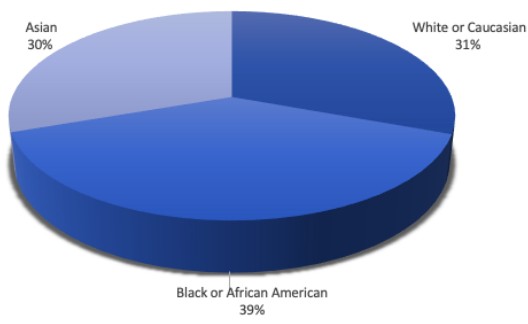

**Figure 13.** The occurrence of glaucoma based on race category.

To provide a broader perspective on the environmental and socio-economic factors that may influence the health status of patients, we have also included a statistic regarding the marital status of the individuals involved. It should be noted, however, that from a medical perspective, the occurrence of glaucoma is not directly associated with marital status. The dataset contains five categories of marital status: Married/Civil Union or Life Partner, Single, Divorced, Widowed, Legally Separated, and Unknown. Figure 14 shows a distribution of graphs according to gender and the mentioned marital statuses.

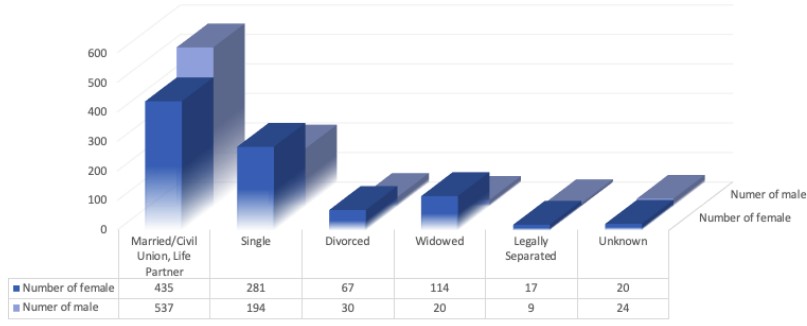

**Figure 14.** The occurrence of glaucoma based on gender and marital status category.

Considering that in this dataset, individuals from the first category—Married/Civil Union or Life Partners are overrepresented, according to the conducted statistics the highest percentage of individuals with glaucoma—56% belong to the aforementioned category, followed by the "Single" category—27%. Conversely, the fewest cases of glaucoma were identified among those in the categories Legally Separated—1%, Unknown—2%. All these aspects are highlighted in Figure 15.

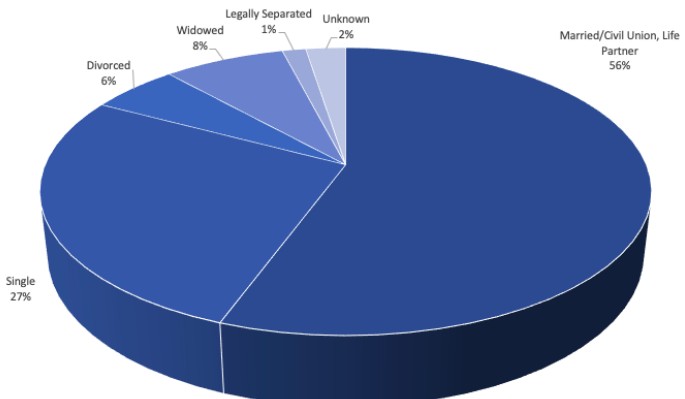

**Figure 15.** The occurrence of glaucoma based on marital status category.

As a result of the analysis presented earlier, the dataset could be used as a starting point for building a Digital Twin associated with the patient. To ensure that the Digital Twin provides results with a high degree of accuracy, the dataset should be enhanced by incorporating information obtained from ophthalmological consultations, as well as environmental factors that may influence the patient's health, predisposing them to the onset of glaucoma. All the information contained in the dataset can be integrated for the purpose of building predictive models for diagnosing glaucoma, thus contributing to the construction of a Digital Twin associated with the patient.

In conclusion, concerning the advancement of predictive medicine and the construction of Digital Twins for glaucoma identification with the aim of providing personalized treatment options in the future, a range of demographic parameters need to be considered. Depending on the results obtained from the interpretation of the previously conducted statistics, correlations were identified in terms of demographic parameters, which help both in segmenting patients into relevant categories and in proposing personalized treatments. Thus, the highest number of glaucoma cases occurs in patients aged over 50 years and in those who are part of the "Black or African American" race, with gender or marital status not being the main determining factors. As a result of factors indicating an increased risk of glaucoma, measures can be taken to help improve the effectiveness of personalized treatment and the quality of life of patients. Based on these correlations, Digital Twins

can be built to assist in preventing or diagnosing glaucoma in its early stages, simulating personalized treatment options and patient reactions.

*5.2. Training the Neural Network for Glaucoma Detection*

Utilizing input data consisting of RNFLT maps, their corresponding visual field mean deviation (MD), and the indication of glaucoma presence or absence, we constructed a mixed-data network with multiple inputs. The associated model diagram was presented in the Materials and Methods section. The Harvard Glaucoma Detection dataset with 500 samples (Harvard-GD500) was employed to train the model [36,37].

According to Lokhande et al. [39], the processing of ophthalmic images for glaucoma diagnosis is notably complex. The anatomical details of the eye, such as retinal thickness, contribute significantly to this complexity. Based on the information presented in [39], the glaucoma detection process was carried out using conventional Machine Learning methods: classification and regression. One of the parameters contributing to the diagnosis of glaucoma involves the use of retinal nerve fiber layer (RNFL) thickness maps, derived from OCT images, to identify corneal thickness. Also, with the help of Machine Learning algorithms, biomarkers for thinning and atrophy of the optic nerve are extracted. Therefore, detecting the thinning of nerve fibers may indicate the progressive deterioration of retinal nerve fiber layer thickness (RNFLT), a significant indicator of glaucoma presence. Textural features provide information regarding the distribution of retinal nerve fibers, which can be extracted using texture analysis methods. With the assistance of the specialist doctor who will contribute through the feedback system to train the Digital Twin, morphological characteristics such as length, curvature, and direction of nerve fibers can be provided to diagnose both glaucoma and predisposition to this pathology. Based on an enhanced dataset, both the mean and standard deviation will be considered to construct a model of distribution for the intensity and thickness of retinal nerve fibers. However, for the realization of this work, the emphasis has been placed on introducing regularization based on contrastive learning, which leads to the appropriate representation of images in the form of clusters, as well as adequate differentiation based on the feedback received from the specialist doctor. Thus, vector representations are obtained that highlight the similarities and differences in images. This approach integrates the doctor's opinion into the prediction process, ensuring convergence towards a robust method regarding the analysis and interpretation of images for glaucoma identification.

Processing RNFLT maps results in Figure 16a indicating the presence of glaucoma, while Figure 16b denotes the absence of glaucomatous status.

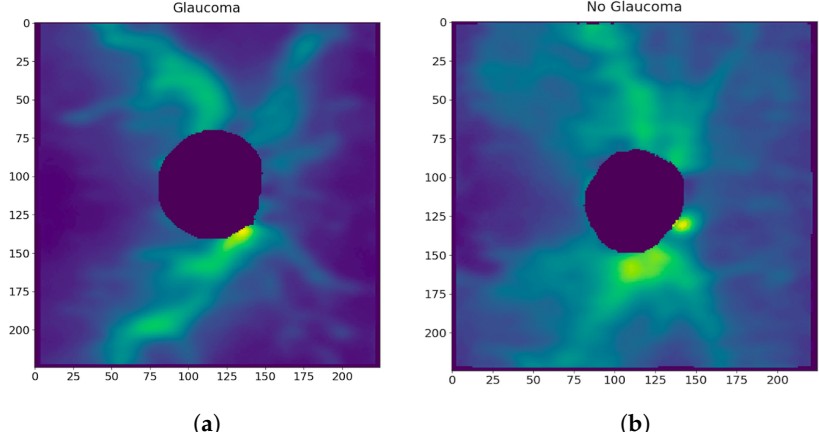

| (**a**) | (**b**) |

**Figure 16.** The processing of RNFLT maps—glaucoma or non-glaucoma status. (**a**) Glaucoma, (**b**) Non-glaucoma.

After training the model, a 5 × 5 image was created, where each subplot represents an image from the dataset—Figure 17. Labels were also added to identify the presence of

pathology: 1 indicates that the patient was diagnosed with glaucoma, while 0 indicates that the patient does not suffer from glaucoma.

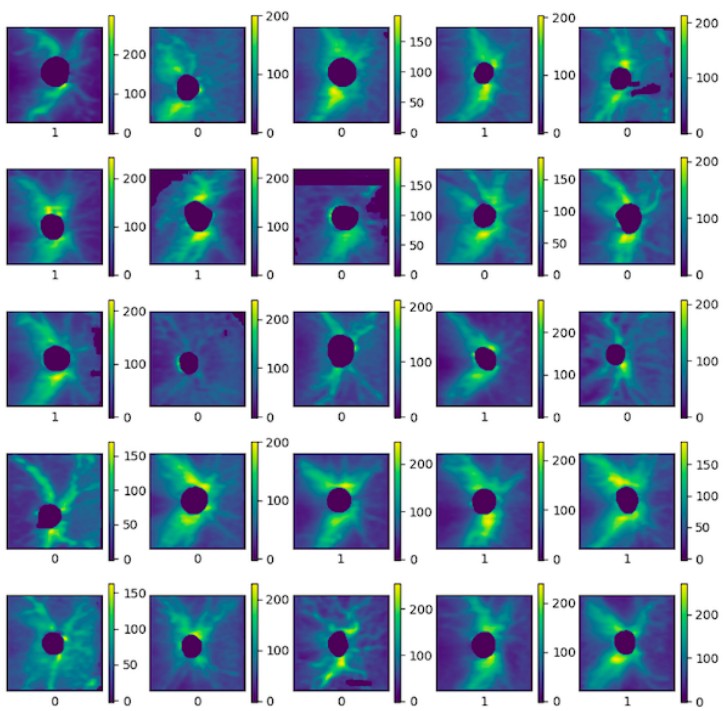

**Figure 17.** The presence of pathology—5 × 5.

After loading the trained model, its performance was evaluated on a test dataset, with 10% of the data allocated for this purpose. Upon loading and normalizing the data, the trained model was then applied to the test dataset to make predictions. Subsequently, the accuracy of these predictions was calculated.

The final step involved creating graphs to visualize the evolution of the loss and accuracy of the model during training. The graph for the loss function was generated using data from both the training and test datasets—Figure 18a. Similarly, the accuracy graph utilized data from both the training and test datasets—Figure 18b. An accuracy of 84% was achieved on the test data.

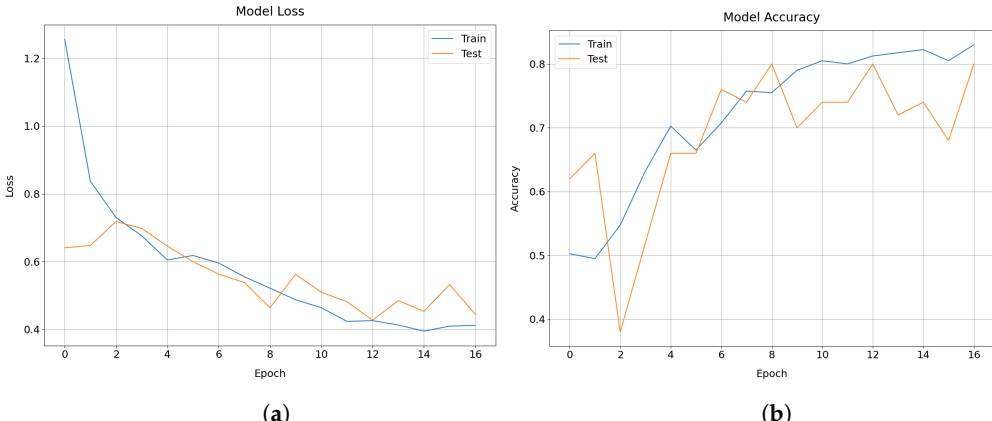

(**a**) (**b**)

**Figure 18.** The evolution of the loss and accuracy of the model during training. (**a**) The evolution of the loss, (**b**) The accuracy of the predictions.

As a result of the obtained results, the accuracy achieved does not fully correspond to expectations, leading to adjustments regarding the methods used. To improve the results, considerations will be made for optimizing the neural network architecture, adjusting

parameters, as well as using new techniques. As mentioned earlier, the dataset used needs to be improved to include a diverse range of demographic data from patients, as well as measurements obtained during ophthalmic consultations, in order to construct not only a Digital Twin associated with the patient but also a pathology-associated one—a Digital Twin for glaucoma identification.

The applicability of the Digital Twin in ophthalmology, especially in glaucoma detection, holds considerable potential, even though it currently represents only a concept. Thus, by integrating each patient's data, and medical history, and employing predictive modeling, one can discuss shaping personalized treatment plans, simulating surgical procedures, continuously monitoring patient health, and optimizing the diagnostic process. The Digital Twin can be considered a paradigm that emphasizes revolutionizing the entire medical system, with a focus on patient health. The use of Digital Twin in this field can lead to changing the way medical staff interacts with patient clinical data, integrating technology with medical expertise. All of these contribute to transforming how patient conditions are diagnosed, treated, and managed, shaping a symbiosis between science and technology.

## 6. Discussion

As mentioned in the Section 5, the accuracy achieved does not meet our expectations. One reason for this may be the architecture and parameters of the network, leading to a mismatch between the model's complexity and the availability of training data. Also, the quality of the dataset used to train and test the model should be taken into consideration. In the initial phase of constructing the Digital Twin to obtain conclusive results that ensure performance and the construction of a general model, the dataset used must be comprehensive enough to highlight the variations encountered in medical practice. Another argument could be the use of inadequate techniques in the data preprocessing and model regularization stages, which led to the disruption of the noise removal process. Last but not least, another factor could be the use of algorithms that are not specific enough to identify subtle signs of the pathology and to avoid diagnostic errors.

In the following, some future research directions regarding the application of Digital Twin in ophthalmology will be presented. Based on the interpretation of the previously obtained statistics, it is desired to construct Digital Twins corresponding to different categories of individuals, taking into account genetic predisposition, integrating information obtained from ophthalmological consultations, creating personalized treatments, and simulating patient reactions.

The next stage emphasizes the adoption of a multi-model approach to ensure the interoperability of the entire system, with a focus on using behavioral modeling based on discrete event systems. From this perspective, a modeling of the cell cycle could be performed, and based on the interactions at the molecular level, the patient's predispositions to certain pathologies and their diagnosis could be identified.

Another application for the use of discrete event systems is represented by the simulation of the impact of personalized treatments, thereby providing a comprehensive view of the patient's symptoms as a result of the administration of the treatment. Also, the events can represent various stages of the pathologies such as regression, stagnation, and evolution, a fact that would help the medical staff in making decisions.

## 7. Conclusions

The creation of Digital Twin models by integrating specific elements of Systems Medicine and Predictive Medicine helps identify and manage eye diseases, implicitly improving diagnosis. Thus, using Digital Twin models, doctors can simulate the behavior of the eye in different situations, which helps them quickly and efficiently identify potential eye problems. Predictive Medicine and Machine Learning offer a holistic approach to simulating scenarios of various eye pathologies and offering personalized solutions for patients.

The use of artificial intelligence in constructing a Digital Twin associated with the patient for glaucoma detection not only provides recommendations for subsequent interventions such as recommending a doctor's visit or regular monitoring of parameters influencing the progression or stagnation of the pathology. Artificial intelligence contributes to the analysis and interpretation of clinical data, providing possible progression options for glaucoma and simulating the patient's reactions to the proposed medication.

A structured architecture with five layers has been proposed with the aim of building a Digital Twin associated with the patient for glaucoma identification. As part of this architecture, statistics were conducted to identify correlations based on various demographic parameters, such as age, gender, and race that favor the onset of glaucoma. Additionally, the results obtained from training the neural network for glaucoma identification were presented. These results, Section 5, did not meet the expectations, motivating in Section 6 the changes we intend to make to improve the model.

According to the literature that we consulted, a predictive model-based Digital Twin is not yet sufficiently approached and applied in ophthalmology. That is why we consider that the results of our work presented in this paper represent a novelty in personalized and predictive ophthalmology.

**Author Contributions:** Conceptualization, M.-E.I.; Methodology, S.-I.C. and M.-E.I.; Validation, M.-A.M.; Formal analysis, S.-I.C.; Investigation, D.-I.C. and M.-E.I.; Resources, A.C.; Data curation, E.P.; Writing—original draft, M.-E.I.; Writing— review & editing, E.P., M.-E.I. and D.-I.C.; Visualization, T.-C.M.; Supervision, M.-A.M., S.-I.C. and T.-C.M. All authors have read and agreed to the published version of the manuscript.

**Funding:** This research received no external funding.

**Institutional Review Board Statement:** Not applicable.

**Informed Consent Statement:** Not applicable.

**Data Availability Statement:** The data is publicly available.

**Conflicts of Interest:** The authors declare no conflict of interest.

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
