# Peer review of "Digital Twin Models for Personalised and Predictive Medicine in Ophthalmology"

_technologies, doi:10.3390/technologies12040055_

Round 1
Reviewer 1 Report
Comments and Suggestions for Authors
This paper researches a digital twin models for personalized and predictive medicine in ophthalmology, give an architectural framework for digital twin, risks and vulnerabilities. The research of this paper has some significance, but there are many shortcomings in this paper, which are as follows:
(1) The specific principle is not clearly explained by the author about digital twin models for personalized and predictive medicine. The author only gives a general description, and it is suggested that the author give a detailed principle and method according to the research object.
(2) in part 3.2, the authors describe the vulnerabilities at the digital twin level, how to improve the method? What are the specific measures?
(3) in part 5 and 6, although the author gives some analysis and results, there is no corresponding model in the method section.
(4) The author introduces a deep learning method, but does not give a specific model, and suggests that the author give a suitable model that conforms to the method proposed in this paper.
(5) This paper lacks comparative validation of research methods.
Comments on the Quality of English LanguageThe expression of the paper is relatively good, but the description of some sentence needs further improvement
Reviewer 2 Report
Comments and Suggestions for Authors
Please, refere to the attached document for comments.

Round 2
Reviewer 2 Report
Comments and Suggestions for Authors
Thank you for considering the remarks and integrating them in extensive re-work.